# Virological non-suppression among adult males attending HIV care services in the fishing communities in Bulisa district, Uganda

**Ignatius Senteza** [1]*, **Barbara Castelnuovo**[2], **David Mukunya**[3], **Fredrick Makumbi**[1]

**1** Department of Epidemiology and Biostatistics, School of Public Health, College of Health Sciences, Makerere University, Kampala, Uganda, **2** Research Department, Infectious Diseases Institute, Makerere University, Kampala, Uganda, **3** Department of Community and Public Health, Faculty of Health Sciences, Busitema University, Busitema, Uganda

☯ These authors contributed equally to this work.

\* senteza28@gmail.com

## Abstract

### Background

Virological non-suppression is a critical factor in driving HIV transmission rates, yet there is limited data available on the determinants of this phenomenon, particularly in fishing communities where the incidence of HIV is disproportionately high. We aimed to determine the prevalence and determinants of virological non-suppression among adult males (≥15 years) attending HIV care services in the fishing communities of Bulisa district.

### Methods

We conducted a cross-sectional study among all adult males (≥15 years) living with HIV who were resident within the fishing communities, and in care for atleast 6 months at the six health facilities offering HIV services in the fishing communities in Bulisa district. To obtain data on patient and health facility characteristics, we reviewed patients' records and conducted face-to-face interviews with the participants. We conducted descriptive and regression analyses using modified Poisson regression, accounting for data correlation of observation at the facility level to obtain prevalence ratios (PR) with 95% confidence intervals in Stata version 14.0.

### Results

413 participants were studied and 379 (91.8%) were interviewed. The participant's average age (SD) was 40 (10.7) years and 70.5% (267/379) were engaged in the fishing business. The prevalence of virological non-suppression was 88/413–21.3% (95% CI: 18%-26%). Factors associated with higher odds of virological non-suppression included: Age 26–50 years (adj.PR = 1.53, 95%CI: 1.11–2.08) and 15–25 years (adj.PR = 2.99, 95%CI: 1.27–7.05) compared to age above 50 years; unemployment (adj.PR = 1.28, 95%CI: 1.10–1.49); hazardous use of alcohol (adj.PR = 1.34, 95%CI: 1.10–1.62); non-mobility between fish

**Funding:** SI was partly funded by the Gilead foundation. BC was partly funded by the Fogarty International Centre, National Institute of Health (grant# 2D43TW009771-06 "HIV and co-infections in Uganda"). Other funds were mobilised from personal savings. The management of people living with HIV was partly supported by the President's Emergency Plan for AIDS Relief through the United States Centres for Disease Control (CDC) and Prevention and the terms of a cooperative agreement number NU2GGH001294-03-05". There was no additional external funding received for this study. The funders had no role in study design, data collection and analysis, the decision to publish, or the preparation of the manuscript.

**Competing interests:** The authors have declared that no competing interests exist.

landing sites (adj.PR = 1.37, 95%CI: 1.003–1.87); distant HIV treatment services (adj.PR = 1.37, 95%CI: 1.11–1.69) and TB diagnosis (adj.PR = 1.87, 95%CI: 1.33–2.64).

## Conclusion

Virological non-suppression among people living with HIV in fishing communities along the shores of Lake Albert is alarmingly high, exceeding the UNAIDS threshold of 10% by two-fold. Several key determinants were identified, including hazardous alcohol use, unemployment, and access barriers to HIV treatment services.

## Introduction

The World Health Organization (WHO) identifies key populations as groups that are particularly vulnerable to contracting Human Immunodeficiency Virus (HIV) [1]. In Africa, fishing communities represent a significant proportion of these at-risk populations [2, 3] due to the high prevalence of HIV within these communities [4]. Specifically, along the shores of Lake Victoria (the biggest lake in Uganda), the prevalence of HIV is as high as 19.7% [5] which is three times higher than the general population's prevalence of (6.2%) [6].

The fishing business is often characterized by male dominance, high-risk sexual behaviours [6], frequent alcohol consumption [7] and mobility between different fishing sites [8]. These factors contribute to the potential transmission of HIV, highlighting the need for more effective methods of prevention. One such approach is recognizing the importance of undetectable viral loads in achieving untransmittable HIV status [9–12]. To better understand the factors that contribute to virological suppression among individuals with HIV, it is important to contextualize these determinants. By gaining a better understanding of the factors that contribute to virological suppression among individuals with HIV, we may be able to reduce the incidence of HIV in fishing communities [13, 14].

Bulisa district, located along the shores of Lake Albert, has a significant portion of its population (14.2%) engaged in the fishing business [15]. While the prevalence of HIV among these fishing communities is not known, it is likely to be higher than that of the general population due to the high rates of HIV observed in fishing communities [5]. However, the overall prevalence of HIV in Bulisa district, based on 2020 program data in DHIS-2, was lower at 5.1% [16] as compared to the national prevalence of 6.2% reported in the UPHIA report of 2017 [6].

A study carried out in Uganda using national programme data in 2017 found that the prevalence of virological non-suppression among males with HIV was 13%. However, in Bulisa district during the same time, the prevalence of virological non-suppression amongst all adult males for the same period was much higher, at 20% [17]. This is particularly concerning, as it exceeds the UNAIDS virological non-suppression threshold of 5% by 2030 [18] and twice as high as the UNAIDS threshold of 10% that was supposed to be achieved by 2020 [19].

While there are program data available on virological non-suppression amongst PLHIV at district level in Uganda, little is known about the burden of virological non-suppression specifically among the male-dominated fishing communities. Furthermore, the association of virological non-suppression and key characteristics prevalent in fishing communities such as risky sexual behaviours [20], hazardous alcohol consumption [21] and high mobility [8] is not well understood. Although certain factors have been linked to virological non-suppression in various population groups, these may not apply to fishing communities.

This study aimed to determine the prevalence of virological non-suppression and the associated factors among adult males (15+) living with HIV in the fishing communities of Bulisa district.

## Methodology

### Study setting

The study was conducted in the fishing communities situated along the shores of Lake Albert in Bulisa district, which is approximately 288.5km west of Kampala, the capital city of Uganda [15]. The estimated population according to the National Population and Housing Census 2014 is 113,161 [22]. According to district reports, around 2000 (1.76%) individuals are living with HIV in these communities, with males constituting 45% of the population [16]. Notably, a majority of those with HIV (approximately two-thirds) reside in fishing communities [16]. Fishing activities typically occur at fish landing sites where boats dock, and the trading of fish takes place.

Fishermen often move between fish landing sites, depending on seasonal changes, to increase their chances of catching fish. There are seven public health facilities in the district, but only six are easily accessible to ten or more fishing communities. These facilities routinely assess the virological status of adult patients under HIV care six months after initiating ART treatment and every 12 months thereafter [23].

Patients who have virological non-suppression are typically provided with three intensive adherence counselling sessions before undergoing a repeat viral load test. The management of patients living with HIV is based on several differentiated service delivery models which depend on their degree of stability. For instance, patients who are virologically suppressed, are in WHO clinical stage I or II, have no active TB disease or have completed the intensive phase of TB treatment are considered stable [23]. Stable patients have access to different treatment modalities, including the fast-track drug refill model where patients who do not have any complaints can receive drugs without seeing a clinician. Additionally, there's the community drug refill model where a patient can obtain their medication from the community. Lastly, the client-led ART delivery model is available, whereby patients form small groups and take turns to collect and deliver their drugs. Patients who are deemed unstable are typically enrolled on the facility-based improvement modality which involves closer monitoring and management of individuals with serious illnesses and advanced HIV disease is considered [23]. In Bulisa district, approximately 47% of adult males living with HIV are enrolled on to the fast-track refill model while 41% are on the facility-based management model. A smaller number of patents, approximately 6%, are enrolled on the client-led ART delivery model, and 6% are equally distributed between facility-based groups and community drug distribution models [16].

### Study design

To determine the prevalence of virological non-suppression and factors associated with it among adult males (aged 15 years and above) living with HIV in the fishing communities of Bulisa district, we conducted a cross-sectional study involving patient records and interviews.

### Participants selection

Our study involved all adult males (aged 15 years and above) living with HIV and residents of the fishing communities in Bulisa district. The participants were required to have received ART treatment for at least 6 months between January 2019 and January 2020. We screened patient records from the ART registers and/ or the electronic medical records system to obtain

participants' demographics, duration on ART and place of residence. Those who met the inclusion criteria were documented on a master list using their treatment numbers and facility code.

To assess the contribution of the health facility to virological non-suppression, we conducted face-to-face interviews with the HIV clinic nurse/ clinical officer in charge at each of the six health facilities serving patients living with HIV in the fishing communities. The selection was done purposively.

## Data collection and management

To obtain data for our study, we used a combination of patients' files and primary data collection through face-to-face interviews of patients and HIV clinic charges using questionnaires and data abstraction tools. The patients' questionnaire was structured and designed based on characteristics common among residents of fishing communities and factors previously associated with virological non-suppression from other population segments [17, 21, 24–33].

The Health worker's questionnaire was structured and designed based on factors identified in previous qualitative and quantitative studies that focused on the quality of health care [34–41]. Data collection was carried out by trained research assistants and questionnaires were translated into Lugungu (the native language).

To minimize non-responsiveness, we ensured the completeness of the data collection tools and followed up on missed opportunities using phone calls and physical follow-ups.

Our outcome of interest was virological non-suppression, which we defined as a viral load of less than 1000 copies/ml. Only up-to-date viral load results were considered and we defined an up-to-date viral load result as a result obtained within 12 months from the time of taking off the blood sample at the time of data collection. The independent variables included characteristics salient among residents of fishing communities and those previously associated with virological non-suppression from other studies. These included: hazardous use of alcohol which was assessed using the 12 points AUDIT C tool where a score $\geq 4$ indicated hazardous use of alcohol [42], mobility based on the frequency of movement between fishing landing sites, sexual behaviour assessed based on the number of sexual partners and condom use within 12 months. Occupation was categorised as unemployed, involved in the fishing business or other jobs [8, 20, 21, 43]. Other jobs included farming, transport and general trade, while the fishing business included active fishing and trading in fish.

Patient-related characteristics previously associated with virological non-suppression included age measured in complete years (15–25, 26–50, >50 years); marital status (single, divorced, married, cohabiting and widowed); ART regimen; line of treatment (first, second, third line); ART side effects and how they disrupted work, frequency of ART regimen (once and twice daily); adherence based on pill count by the attending clinician categorized as good (>95%), fair (80–95%), poor (<80%); fulfilment of clinical appointments; disclosure of HIV status (one option from a list of provided options); belongingness to a treatment support group; duration on ART measured in months; daily average income measured in Ugandan shillings (I USD = UGX 3500) categorised according to income level and TB diagnosis with 12 months [17, 21, 24–33]. Health facility characteristics included: the level of the health facility (HC II, HC III, HC IV and hospital levels with the HCII level being the lowest point of care and the hospital being the highest point of care within the district; perceived length of appointment intervals (long, short, neither long or short) and the perception whether HIV services were extended closer to the workplace/home; perceived quality of health education and the quality counselling. These characteristics were selected based on previous studies [34–41]. All data were captured using entry screens designed with EPInfo software and then exported into Microsoft Excel.

## Data analysis

Data were analysed using Stata version 14.0 and summarized using frequencies and proportions. At the exploratory stage, we realised that 34 ($<$10%) participants with the outcome of interest were missing data from face-to-face interviews. These included those who had transferred out, died, were lost to follow up or were displaced due to floods along Lake Albert. Descriptive analyses were performed and results were reported using frequencies and proportions.

Age (years) was normally distributed, and therefore, mean (SD) was used as a measure of central tendency. However, the duration on ART (months) was skewed, and thus, median (IQR) was used as the measure of central tendency. The prevalence of virological non-suppression was the proportion of participants with an up-to-date viral load who had virological non-suppression.

During regression analysis, participants with missing data were excluded and a complete case analysis was considered. We used bivariable analysis to obtain the strength of association between each independent variable with virological non-suppression. Poisson regression was used to generate crude prevalence ratios (CPR) with 95% confidence intervals and p-values. Multicollinearity was assessed using variance inflation factor (VIF = 1/ (1-R2)) where R is the correlation coefficient. A VIF greater than or equal to 10 indicated serious multicollinearity [44–46]. Preference of factors used at the multivariable analysis level was based on: biological plausibility; *p*-value less than 0.25 at bivariable analysis [47] and uniqueness of the factor (s) to the fishing community. These criteria were used to ensure that the most relevant and significant factors were included in the analysis.

At the multivariable analysis level, backward elimination modelling was conducted using modified Poisson regression accounting for data correlation of observation at facility level to obtain adjusted prevalence ratio (adj.PR) as measures of association, with corresponding 95% confidence intervals and p values. Health workers' responses from face-to-face interviews were analysed using frequencies, and results were summarized using tables.

## Ethical consideration

We sought approval from the Research and Ethics Committee of Makerere University—School of Public Health to carry out the research. Written informed consent was obtained from all study participants, and assent was obtained from guardians of participants aged 15 to 18 years.

## Results

Fig 1 shows the description of the data that were used in the study.

Out of the 829 adult males receiving HIV care in Bulisa district between January 2019 and January 2020, 462 (55.7%) fulfilled the eligibility criteria, and 413 (89.4%) of eligible participants had an up-to-date viral load.

Out of 829 participants line listed, 462 fulfilled the inclusion criteria of whom 413 had an up-to-date viral load test result. 367 were excluded because they were non-residents in the fishing communities or had been enrolled on ART for less than 6 months and were not due for viral load monitoring according to the guidelines for viral load monitoring in Uganda. The response rate for the face-to-face interviews was 379/413 (91.8%). The 34 (8.2%) participants who were not interviewed had either died, were lost from care, transferred out or displaced due to floods.

The average (SD) duration on ART (months) was comparable between participants who were interviewed [41.7 (29.9)] and those who were not interviewed [40.3(30.4)], p-

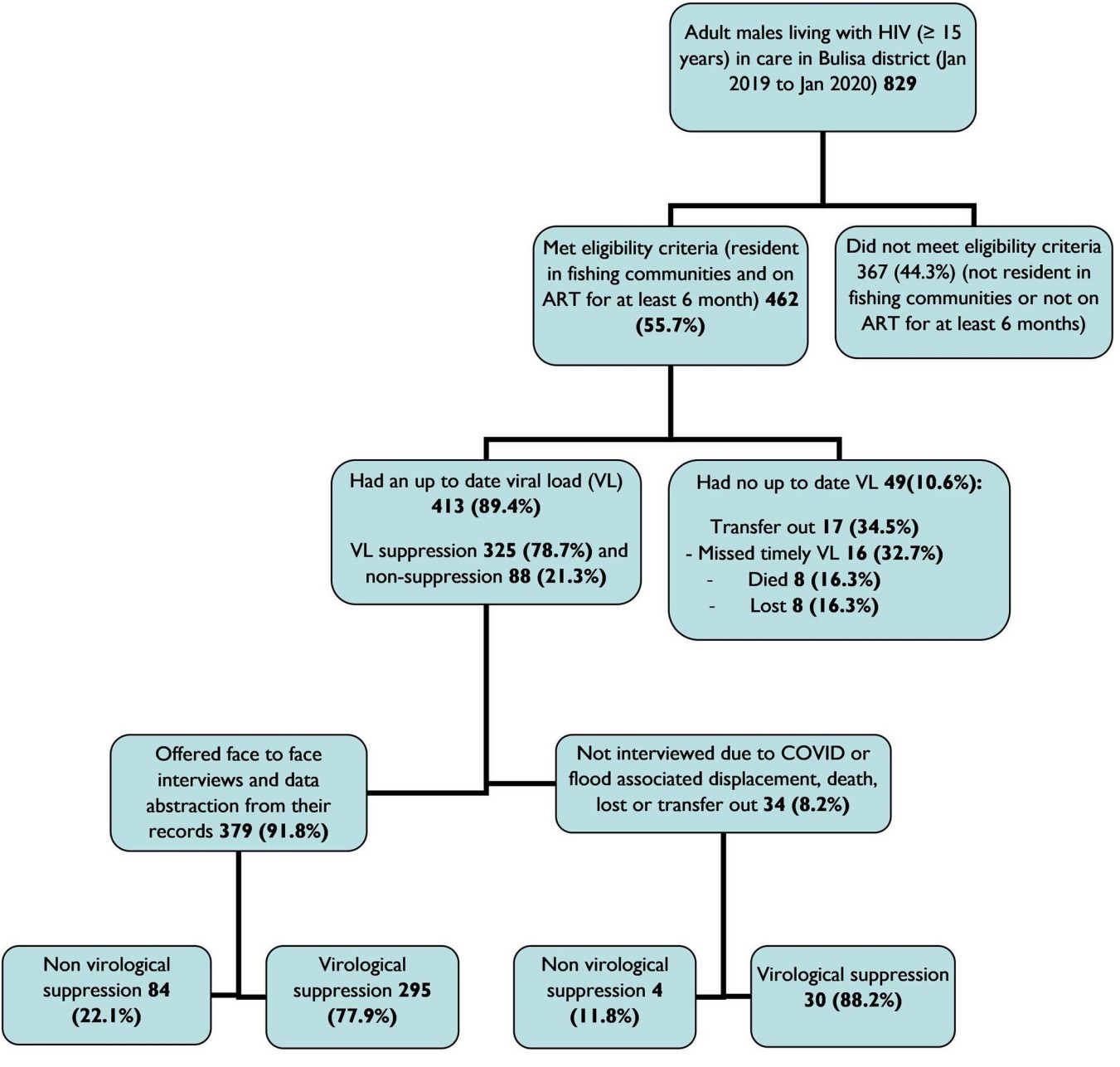

**Fig 1. A consort flow diagram showing data that were used.**

value = 0.794. However, those interviewed and not interviewed differed in their mean (SD) age (years), 40 (10.7) versus 35.9 (9.3), p = 0.011.

## Demographic, individual and health facility-related characteristics of the participants

Table 1 shows the demographic, individual and health facility-related characteristics of the participants.

**Table 1. Demographic, individual and health facility-related characteristics of participants n = 413.**

| Characteristics | Frequency (n) | Percentage (%) |
|---|---|---|
| Age category | | |
| 15 to 25 | 19 | 5.0 |
| 26 to 50 | 291 | 76.8 |
| > 50 years | 69 | 18.2 |
| **Marital status** | | |
| Single | 70 | 18.5 |
| Divorced | 52 | 13.7 |
| Married | 113 | 29.8 |
| Cohabiting | 133 | 35.1 |
| Widowed | 11 | 2.9 |
| **Occupation** | | |
| Others jobs | 90 | 23.8 |
| Engaged in fishing business | 267 | 70.5 |
| Not employed | 22 | 5.8 |
| **Average income** | | |
| Less than UGX 10,000 | 226 | 59.6 |
| Between 10,000–20,000 | 133 | 35.1 |
| More than UGX 20,000 | 20 | 5.3 |
| **Up-to-date viral load** | | |
| Suppression | 325 | 78.7 |
| Non-suppression | 88 | 21.3 |
| **Ever changed regimen from baseline** | | |
| No | 107 | 26.0 |
| yes | 306 | 74.0 |
| **Line of the current regimen** | | |
| First | 371 | 89.8 |
| Second/ third | 42 | 10.2 |
| **Duration on ART** | | |
| Less than 24 months | 141 | 34.1 |
| 24–50 months | 129 | 31.2 |
| Greater than 50 months | 143 | 34.6 |
| **Documented adherence level** | | |
| Good | 312 | 75.5 |
| Fair | 60 | 14.5 |
| Poor | 41 | 10.0 |
| **Number of sexual partners in the previous 12 months** | | |
| None | 21 | 5.6 |
| One | 174 | 46.3 |
| More than one | 181 | 48.1 |
| **Partner HIV status** | | |
| Don't know | 105 | 30.9 |
| Negative | 90 | 26.5 |
| Positive | 145 | 42.7 |
| **Condom use** | | |
| Never | 157 | 44.2 |
| Once in a while | 165 | 46.5 |
| Every time | 33 | 9.3 |

(*Continued*)

**Table 1.** (Continued)

| Characteristics | Frequency (n) | Percentage (%) |
|---|---|---|
| **Missed taking ART in the last 12 months** | | |
| No | 150 | 40.0 |
| yes | 228 | 60.0 |
| **HIV disclosure** | | |
| No one | 31 | 8.2 |
| Any other family member | 131 | 34.7 |
| Workmate/ friend/neighbour | 37 | 9.8 |
| Wife | 179 | 47.4 |
| **Hazardous use of alcohol** | | |
| Non-hazardous use | 261 | 63.2 |
| Hazardous use | 152 | 36.8 |
| **Frequency of moving between fish landing sites in a year** | | |
| Never | 156 | 41.5 |
| Once or twice | 149 | 39.6 |
| More than twice | 71 | 18.9 |
| **ARVs and stability at work** | | |
| No disruption | 271 | 71.9 |
| Disrupts work | 106 | 28.1 |
| **Permanent resident of Bulisa** | | |
| No | 82 | 21.7 |
| Yes | 296 | 78.2 |
| **Participant's facility level for treatment** | | |
| Health Centre II level | 24 | 5.81 |
| Health Centre III level | 254 | 61.50 |
| Health Centre IV level | 101 | 24.46 |
| Hospital level | 34 | 8.23 |
| **Belonging to a treatment support group** | | |
| No | 255 | 67.8 |
| Yes | 121 | 32.2 |
| **Assessment of Confidentiality at the health facility** | | |
| No Confidentiality | 98 | 26.0 |
| Some confidentiality | 93 | 24.7 |
| Maximum confidentiality | 186 | 49.3 |
| **Quality of health education talks** | | |
| Do not happen at all | 19 | 5.0 |
| Less interactive and rushed | 67 | 17.7 |
| Interactive | 293 | 77.3 |
| **Perceived length of clinic appointments** | | |
| Long | 49 | 12.9 |
| Neither long nor Short | 240 | 63.3 |
| Short | 90 | 23.8 |
| **Viral load turnaround time** | | |
| Did not know | 34 | 9.0 |
| 1 to 2 months | 207 | 55.8 |
| More than 2 months | 20 | 5.4 |
| Less than 1 month | 110 | 29.7 |
| **HIV services extended near work/home** | | |

(*Continued*)

**Table 1.** (Continued)

| Characteristics | Frequency (n) | Percentage (%) |
|---|---|---|
| Yes | 210 | 55.6 |
| No | 168 | 44.4 |
| Current ART regimen | | |
| TDF/3TC/EFV | 57 | 15.4 |
| TDF/3TC/DTG | 281 | 74.1 |
| TDF/3TC/ATV/r | 16 | 4.2 |
| AZT/3TC/NVP | 9 | 2.4 |
| Other second-line regimens | 13 | 3.4 |
| Other first-line regimens | 3 | 0.8 |
| **Diagnosed with TB within 12 months** | | |
| No | 356 | 93.9 |
| Yes | 23 | 6.1 |

The mean (SD) age was 40 (10.7) years and the median (IQR) duration on ART was 37 (17–57) months. The proportion of participants with virological non-suppression was 21.3%. Three-quarters of the participants were taking TDF/3TC/DTG as their current ART regimen and more than two-thirds were engaged in the fishing business. Only 5% of participants reported earning more than UGX 20,000 (about 6 US dollars) per day on average.

About one-third of the participants reported having had 2 to 3 sexual partners in the previous 12 months. One-third reported never knowing their partner's HIV status. Half of the participants had partners who were living with HIV. About one-third were engaged in hazardous alcohol use and one-half of them moved between fish landing sites at least once in the past 12 months. More than three-quarters of the participants were permanent residents of Bulisa district and about two-thirds sought their treatment from mid-level health facilities (either HC III or HC IV).

The prevalence of virological non-suppression amongst adult males (≥15 years) living with HIV in the fishing communities of Bulisa was 22.2% (95% CI: 18% -26%).

**Factors associated with virological non-suppression.** Table 2 summarises the results from the bivariable and multivariable analyses.

*At bivariable analysis.* Age between 15 to 25 years was associated with a higher prevalence of virological non-suppression (CPR = 2.82, 95% CI = 1.05–7.58) compared to participants greater than 50 years. The prevalence of virological non-suppression amongst participants who disclosed their HIV status to either workmate, friend, or neighbour was 0.21 times (CPR = 0.21, 95% CI = 0.04–0.99) compared to the prevalence of virological non-suppression amongst participants who never disclosed their HIV status. The prevalence of virological non-suppression amongst participants diagnosed with TB within 12 months before the study was 1.86 times (CPR = 1.85, 95% CI = 1.13–2.63) compared to the prevalence of virological non-suppression among participants who were never diagnosed with TB. Other results are summarised in Table 2.

*At multivariable analysis.* Results for factors salient among patients living with HIV in the fishing communities are described below. Other results are summarised in Table 2.

The prevalence of virological non-suppression among unemployed participants was 1.31 times (adj.PR = 1.31, 95%CI: 1.15–1.50) the prevalence of virological non-suppression amongst participants that were engaged in other jobs other than fishing. The prevalence of virological non-suppression amongst adult males who engaged in hazardous use of alcohol was 1.32 times (adj.PR = 1.32, 95%CI: 1.13–1.54) the prevalence of virological non-suppression

**Table 2. Bivariable and multivariable analysis for determining the factors associated with virological non-suppression amongst adult males living with HIV in the fishing communities of Bulisa district.**

| Factor | Virological suppression status | | Bivariable analysis n = 379 | | Multivariable analysis n = 372 | |
|---|---|---|---|---|---|---|
| | Suppressed n (%) | Non-suppressed n (%) | Crude PR (95%CI) | P-value | Adjusted PR (95%CI) | P-value |
| **Age category** | | | | | | |
| Greater than 50 years | 60(86.96) | 9(13.04) | | | | |
| 26 to 50 | 223(76.63) | 68(23.37) | 1.79(0.89–3.59) | **0.100** | 1.63(1.14–2.32) | **0.007** |
| 15 to 25 | 12(63.16) | 7(36.84) | 2.82(1.05–7.58) | **0.039** | 3.26(1.42–7.48) | **0.005** |
| **Occupation** | | | | | | |
| Others jobs | 72(80) | 18(20) | | | | |
| Fishing/trading in fish | 207(77.5) | 60(22.47) | 1.1(0.66–1.92) | 0.665 | 1.09(0.84–1.41) | 0.699 |
| Not employed | 16(72.73) | 6(27.27) | 1.3(0.54–3.44) | 0.511 | 1.31(1.15–1.50) | **<0.001** |
| **Daily average income** | | | | | | |
| > UGX 20,000 | 18(90.0) | 2(10.0) | | | | |
| < UGX 10,000 | 167(73.89) | 59(26.11) | 2.6(0.64–10.68) | 0.182 | | |
| UGX 10,000–20,000 | 110(82.71) | 23(17.29) | 1.73(0.41–7.33) | 0.457 | | |
| **Belonging to a treatment group** | | | | | | |
| Yes | 95(78.51) | 26(21.49) | | | | |
| No | 198(77.65) | 57(22.35) | 1.04(0.65–1.65) | 0.868 | | |
| **HIV disclosure** | | | | | | |
| No one | 23(74.19) | 8(25.81) | | | | |
| Other family members other than the wife | 95(72.52) | 36(27.48) | 1.06(0.49–2.30) | 0.872 | | |
| Workmate/ friend/neighbour | 35(94.59) | 2(5.41) | 0.21(0.04–0.99) | **0.048** | | |
| Wife | 141(78.77) | 38(21.23) | 0.82(0.38–1.76) | 0.616 | | |
| **Hazardous use of alcohol** | | | | | | |
| Non-hazardous use | 179(78.85) | 48(21.15) | | | | |
| Hazardous use | 116(76.32) | 36(23.68) | 1.12(0.73–1.73) | 0.607 | 1.32(1.14–1.54) | **<0.001** |
| **Frequency of moving between fish landing sites** | | | | | | |
| More than twice a year | 56(78.87) | 15(21.13) | | | | |
| Once or twice a year | 116(77.85) | 33(22.15) | 1.04(0.57–1.93) | 0.880 | 1.12(0.88–1.44) | 0.344 |
| Never | 122(78.21) | 34(21.79) | 1.03(0.56–1.89) | 0.920 | 1.27(1.03–1.56) | **0.020** |
| **ARV disruption towards work** | | | | | | |
| No disruption | 218(80.44) | 53(19.56) | | | | |
| Disrupts work | 75(70.75) | 31(29.25) | 1.5(0.96–2.33) | 0.075 | | |
| **Missed taking ART in the last 12 months** | | | | | | |
| No | 126(84) | 24(16) | | | | |
| Yes | 169(74.12) | 59(25.88) | 1.6(1.0–2.6) | **0.047** | | |
| **Number of sexual partners in previous 12 months** | | | | | | |
| None | 17(81) | 4(19) | | | | |
| Only one | 132(75.9) | 42(24.1) | 1.27(0.45 3.53) | 0.651 | 1.50(0.62–3.64) | 0.374 |
| More than one | 144(79.6) | 37(20.4) | 1.07(0.38–3.01) | 0.893 | 1.03(0.45–2.36) | 0.946 |
| **Marital status** | | | | | | |
| Single | 51(72.86) | 19(27.14) | | | | |
| Divorced | 41(78.85) | 11(21.15) | 0.78(0.37–1.64) | 0.511 | | |
| Married | 89(78.76) | 24(21.24) | 0.78(0.43–1.43) | 0.424 | | |
| Cohabiting | 106(79.7) | 27(20.3) | 0.75(0.41–1.35) | 0.332 | | |
| Widowed | 8(72.73) | 3(27.27) | 1.00(0.30–3.4) | 0.994 | | |
| **Line of the current regimen** | | | | | | |
| First | 284(83.78) | 55(16.22) | | | | |

*(Continued)*

**Table 2.** (Continued)

| Factor | Virological suppression status | | Bivariable analysis n = 379 | | Multivariable analysis n = 372 | |
|---|---|---|---|---|---|---|
| | Suppressed n (%) | Non-suppressed n (%) | Crude PR (95%CI) | P-value | Adjusted PR (95%CI) | P-value |
| Second/ third | 11(27.50) | 29(72.50) | 4.5(2.85–7) | **<0.001** | | |
| **Participant's facility of treatment** | | | | | | |
| Health Centre II level | 16(66.7) | 8(33.3) | | | | |
| Health centre III | 207(81.5) | 47(18.5) | 0.56(0.26–1.17) | 0.124 | 0.62 (0.57–0.68) | **<0.001** |
| Health centre IV | 78(77.23) | 23(22.77) | 0.68(0.3–1.53) | 0.353 | 1.03(0.91–1.16) | 0.659 |
| Hospital level | 24(70.59) | 10(29.41) | 0.88(0.35–2.23) | 0.792 | 1.20(1.03–1.40) | **0.019** |
| **Diagnosed with TB in last 12 months** | | | | | | |
| No | 281(78.93) | 75(21.07) | | | | |
| Yes | 14(60.87) | 9(39.13) | 1.86(1.13–2.63) | **<0.001** | 1.98(1.39–2.82) | **<0.001** |
| **HIV services extended close to workplace/ home** | | | | | | |
| Yes | 170(81) | 40(19) | | | | |
| No | 125(74.4) | 43(25.6) | 1.34(0.87 2.07) | 0.179 | 1.38(1.07–1.77) | **0.012** |

amongst participants who never engaged in hazardous use of alcohol. The prevalence of virological non-suppression amongst participants who solely operated at one fishing landing site was 1.27 times (adj.PR = 1.27, 95%CI: 1.03–1.56) compared to the prevalence of virological non-suppression amongst participants who moved more than twice annually between fishing landing sites. The prevalence of virological non-suppression amongst participants who reported that HIV treatment services were not close to their workplaces was 1.38 times (adj. PR = 1.38, 95%CI: 1.07–1.77) the prevalence of virological non-suppression amongst participants that reported that HIV treatment services were extended closer to their workplaces. The prevalence of virological non-suppression amongst participants diagnosed with TB within 12 months before the study was 1.98 times (adj.PR = 1.98, 95%CI: 1.39–2.82) compared to the prevalence of virological non-suppression amongst participants that were not diagnosed with TB within the same period.

**The role of health facilities towards virological non-suppression amongst adult males living with HIV in the fishing communities of Bulisa district.** Table 3 shows the results from face-to-face interviews among health workers that focused on the preparedness of the health facilities to serve people living with HIV and the quality of care offered to them.

At five out of six health facilities, patients were routinely attended to by a clinician except for Bugoigo HC III where an experienced HIV peer leader occasionally attended to patients. Four out of six facilities offered scheduled health education talks for patients and had peers resident within the fishing communities trained to support fellow patients. Five of the six facilities had staff designated to provide adherence counselling however only three had received training in counselling skills. The main method for assessing adherence at all facilities was daily physical counting of pill balances. All facilities received training in the national HIV treatment and prevention guidelines. Patients at five out of six facilities were served through community drug distribution points (CDDPs) and home refills, whereas three facilities also used CCLADs (community client-led ART delivery). Only one facility had a waiting time of no more than 30 minutes. Five of the six health facilities assessed feedback from patients quarterly. All facilities had quality improvement committees aimed at improving the quality of care among patients.

**Table 3. Preparedness of health workers to serve people living with HIV and quality of services offered to people living with HIV.**

| Variable | Freq of response | Type of response | Response per Health Facility | | | | | |
|---|---|---|---|---|---|---|---|---|
| | | | Butiaba | Bugoigo | Biiso | Bulisa Hosp | Bulisa HC | Kigwera |
| whether there are days the HIV clinic remains unattended to by clinicians | 6 | Yes | | X | | | | |
| | | No | X | | X | X | X | X |
| Whether there is a schedule for health education talks | 6 | Yes | X | X | | | X | X |
| | | No | | | X | X | | |
| Availability of peer leaders residing within fishing communities at the HIV clinic | 6 | Yes | X | X | | | X | X |
| | | No | | | X | X | | |
| whether peer leaders have been trained in supporting fellow positive patients | 4 | Yes | X | X | | | X | X |
| presence of staff designated to provide adherence counselling | 6 | Yes | X | | X | X | X | X |
| | | No | | X | | | | |
| whether staff providing counselling have been trained in counselling skills | 5 | Yes | | | X | | X | X |
| | | No | X | | | X | | |
| Whether the facility staff has received training in the latest HIV guidelines | 6 | Yes | X | X | X | X | X | X |
| Frequency of refresher sessions in HIV management | 6 | Annually | | | X | | | |
| | | Bi-annually | | X | | | | |
| | | Quarterly | X | | | X | | |
| | | monthly | | | | | X | X |
| Whether some patients have missed having their viral loads done timely | 6 | Yes | X | X | X | X | X | |
| | | No | | | | | | X |
| Reasons for missing timely assessment of viral load | 6 | Missing appointments | X | | X | X | X | |
| | | Missing results | | X | | | | |
| | | Stockouts | | | X | | | |
| whether there are communication channels for patients to consult | 6 | Yes | X | X | X | X | X | X |
| whether there are mechanisms to extend services nearer to patients | 6 | Yes | X | X | X | X | X | X |
| Mechanisms used to extend services close to the patients | 13 | CCLADS | X | | X | | X | |
| | | CDDPs | X | | X | X | X | X |
| | | Home refills | X | X | | X | X | X |
| Average waiting time | 6 | < 30 minutes | | | | X | | |
| | | 30–60 minutes | | | X | | | X |
| | | 1–2 hours | X | X | | | X | |
| Frequency of assessing feedback about service delivery from patients | 6 | Quarterly | | | X | | | |
| | | Never | | X | | | X | |
| | | Monthly | X | | | X | | X |
| Availability of a quality improvement team to improve service delivery among patients | 6 | Yes | X | X | X | X | X | X |
| The method used to assess adherence | 8 | Health worker judgment | X | | X | | X | |
| | | Daily physical counts | | X | | X | X | X |
| | | Missing appointments | | | X | | | |

## Discussion and conclusion

Our study aimed to evaluate the prevalence of virological non-suppression and the associated factors among adult males living with HIV in the fishing communities of Bulisa district. Our findings revealed that the prevalence of virological non-suppression was remarkably high (21.3%), which was twice the UNAIDS threshold of 10%. Furthermore, we noted that this figure was considerably higher than the prevalence of non-suppression observed in other fisherfolk settings (9%) [48] as well as among the general male population living with HIV in Uganda (13%) [17]. These results suggest that adult males living with HIV in fishing communities could be contributing significantly to the burden of virological non-suppression. This finding may have significant implications for HIV control and prevention programs in these communities, as targeted strategies could be developed to address the identified risk factors and improve virological suppression rates among HIV-positive individuals living in fishing communities.

According to the study's findings, there is a higher likelihood of virological non-suppression among individuals with hazardous alcohol use. Alcohol dependence has been previously associated with poor adherence to ART [21] which in turn leads to an increased risk for virological non-suppression [17]. Therefore, addressing the challenge of hazardous alcohol use could be an effective strategy to improve virological outcomes among people living with HIV.

Results from this study further revealed that individuals under the age of 50 had a higher likelihood of experiencing virological non-suppression. There was also a correlation demonstrated in the study, indicating that the odds of non-suppression increased with decreasing age. Previous studies that focused on the relationship between age and adherence, a major predictor of virological non-suppression, suggested that older individuals may exhibit better adherence to antiretroviral medication [49, 50]. Better adherence with increasing age could potentially explain the lower likelihood of virological non-suppression. Therefore, interventions to improve medication adherence among younger individuals may be necessary to enhance their virological outcomes and achieve better overall health.

This study found that being unemployed was associated with higher odds of experiencing virological non-suppression. These findings are consistent with previous studies that have also identified a link between unemployment and poorer health outcomes like non-adherence to ART and virological non-suppression [51, 52]. Therefore, interventions targeting employment and income generation may be necessary to improve access to health care and treatment adherence, hence the reduced risk of virological non-suppression.

According to this study, remaining stationed at a single fishing site for an entire year as compared to migrating between fishing landing sites in search of better yield from fishing was found to have a slight association with virological non-suppression. This finding contrasts with previous studies which linked extensive migration with virological non-suppression [53]. In fishing communities, people tend to move between fish landing sites based on fluctuations in fish quantities [8]. Individuals who did not change fishing landing sites may have been among the unemployed, who were found to have higher odds of virological non-suppression. Profiling of adult males at fish landing sites in line with their mobility patterns could help in redirecting efforts to address the challenge of virological non suppression.

This study found that a lack of nearby HIV treatment services was associated with higher odds of virological non-suppression. This finding highlights the importance of providing convenient access to treatment services among fishing communities where mobility between work sites certainly poses a challenge to treatment access. On the positive side, all six health facilities in the fishing communities had mechanisms in place to extend HIV treatment services closer to patients that included CDDPs, CCLADs and direct drug delivery [23]. In contrast, some

studies conducted in non-fishing settings did not find any association between the geospatial patterns of treatment sites, differentiation of treatment, and virological non-suppression [54, 55]. Extending services closer to patients could help in reducing the long time spent at some facilities which manifested as a long waiting time of more than 30 minutes at some facilities. Such long waiting times are likely to discourage patients from seeking regular care from the facilities, resulting in poor adherence and poor treatment outcomes like virological non-suppression. Therefore, it's important to consider the working culture of the patients when designing HIV treatment programs to optimise health outcomes.

This study revealed that having a history of TB disease within the past 12 months was associated with greater chances of virological non-suppression, consistent with an earlier nation-wide study among PLHIV in Uganda [17]. This may be due to the immunosuppressive effects of TB on the body [56], which impedes individuals from achieving viral suppression. This finding highlights the importance of screening for TB and providing appropriate treatment to improve the overall health outcomes of people living with HIV.

This study also found that seeking HIV care at the hospital level was associated with greater odds of virological non-suppression. Conversely, receiving treatment from mid-level (HC III and IV) facilities was associated with lower odds of virological non-suppression compared to the lowest-level health facility. This suggests that the referral system in the district may be effective, as patients with advanced HIV disease with a higher likelihood of virological non-suppression are referred from lower levels to facilities with greater resources to provide comprehensive care. These findings highlight the importance of providing appropriate levels of care for patients based on their clinical needs and the role of efficient referral systems in optimizing health outcomes for people living with HIV.

Contrary to other studies, we did not find any association between sexual behaviour [53] and virologically non-suppression.

Results from the health worker interviews showed that the majority of facilities had trained health workers attending to patients. Training could reduce the impact of virological non-suppression as it necessitates informed decisions and builds the confidence of health workers. Previous studies also indicated that health worker training has an impact on patient outcomes [57].

A strength of our study was the use of interviews to supplement the information obtained from secondary data, which enabled the exploration of unique determinants of virological non-suppression not routinely captured among HIV-positive patients seeking care. Additionally, adjusting for the clustering of variables by health facility level during multivariable analysis allowed us to account for data correlation of observation at facility level.

However, some limitations also need to be noted. Some questions required participants to recall key events, which could have introduced an element of recall bias [58]. Social desirability bias could have influenced some participants to respond to questions assessing sensitive topics, such as sexual behaviour, with reservations. Missingness of some secondary data may have led to selection bias. For example, interviewed participants were older than the non-interviewed participants which could have biased the final results since age was identified as a key predictor of virological suppression [59]. Furthermore, the failure to collect information on the models of care used in the health facilities may have resulted in the missing of valuable information that could inform treatment differentiation policies for people living with HIV in fishing communities. Whereas the study assessed disclosure and partner HIV status, it missed out on assessing these with focus on participants that had multiple partners.

In conclusion, our study found that the prevalence of virological non-suppression among people living with HIV along the shores of Lake Albert is alarmingly high, at twice the UNAIDS threshold of 10%. Key determinants of virological non-suppression in fishing communities include hazardous use of alcohol, unemployment, age younger than 50 years, and

distant HIV treatment services. We, therefore, recommend that the responsible authorities establish HIV medicine refill points at fish landing sites to extend access to HIV treatment services. Additionally, targeted interventions should be developed to address hazardous alcohol use and unemployment. By implementing these recommendations, we can work towards reducing the prevalence of virological non-suppression and improving health outcomes for people living with HIV along the shores of Lake Albert and in fishing communities elsewhere.

## Supporting information

**S1 File. Research fund scholarship.**
(PDF)

**S2 File. Location of Bulisa district in Uganda.**
(PDF)

**S3 File. Coded sheet.**
(XLSX)

**S4 File. Code definitions.**
(PDF)

**S5 File. Consent and ascent forms—English version.**
(PDF)

**S6 File. Consent and ascent form translated version.**
(PDF)

**S7 File. Patient questionnaire–English version.**
(PDF)

**S8 File. Patients questionnaire translated version.**
(PDF)

**S9 File. Data abstraction tool.**
(PDF)

**S10 File. Health workers questionnaire.**
(PDF)

**S11 File. Ethical approval.**
(PDF)

**S12 File. Stata output for the final model.**
(TXT)

**S13 File.**
(DOCX)

## Acknowledgments

The authors of the research manuscript express their gratitude to several parties for their contributions to the study. Specifically, they acknowledge the School of Public Health at Makerere University for providing the platform to conduct the study, the capacity-building department at the Infectious Diseases Institute for their mentorship throughout the research process, the people of Bulisa district for their support during data collection and cooperation throughout

the study, and the author's family members for their moral, social, and financial support during the development of the manuscript.

## Author Contributions

**Conceptualization:** Ignatius Senteza, Barbara Castelnuovo, David Mukunya, Fredrick Makumbi.

**Data curation:** Ignatius Senteza, David Mukunya, Fredrick Makumbi.

**Formal analysis:** Ignatius Senteza, David Mukunya, Fredrick Makumbi.

**Funding acquisition:** Ignatius Senteza, Barbara Castelnuovo, Fredrick Makumbi.

**Investigation:** Ignatius Senteza.

**Methodology:** Ignatius Senteza, Barbara Castelnuovo, David Mukunya, Fredrick Makumbi.

**Project administration:** Ignatius Senteza, Barbara Castelnuovo.

**Resources:** Ignatius Senteza.

**Supervision:** Barbara Castelnuovo, David Mukunya, Fredrick Makumbi.

**Validation:** Ignatius Senteza, David Mukunya, Fredrick Makumbi.

**Visualization:** Ignatius Senteza, David Mukunya, Fredrick Makumbi.

**Writing – original draft:** Ignatius Senteza.

**Writing – review & editing:** Ignatius Senteza, Barbara Castelnuovo, David Mukunya, Fredrick Makumbi.

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
