## [Decision Letter · Decision Letter 0]

10 Aug 2022

PONE-D-21-31809Virological non-suppression among adult males attending HIV care services in the fishing communities in Bulisa district, Uganda.PLOS ONE

Dear Dr. Senteza,

Thank you for submitting your manuscript to PLOS ONE. After careful consideration, we feel that it has merit but does not fully meet PLOS ONE’s publication criteria as it currently stands. Therefore, we invite you to submit a revised version of the manuscript that addresses the points raised during the review process.

We look forward to receiving your revised manuscript.

Kind regards,

Joel Msafiri Francis, MD, MS, PhD

Academic Editor

PLOS ONE

Journal Requirements:

4. Thank you for stating in your Funding Statement: "SI was partly funded by the Gilead foundation

BC was partly funded by the Fogarty International Centre, National Institute of Health (grant# 2D43TW009771-06 “HIV and co-infections in Uganda")

The management of people living with HIV was partly supported by the Presidents Emergency Plan for AIDS Relief through the United States Centers for Disease Control (CDC) and Prevention and the terms of cooperative agreement number NU2GGH001294-03-05"

5. Thank you for stating the following financial disclosure: "SI was partly funded by the Gilead foundation

BC was partly funded by the Fogarty International Centre, National Institute of Health (grant# 2D43TW009771-06 “HIV and co-infections in Uganda")

The management of people living with HIV was partly supported by the Presidents Emergency Plan for AIDS Relief through the United States Centers for Disease Control (CDC) and Prevention and the terms of cooperative agreement number NU2GGH001294-03-05"

Please state what role the funders took in the study.  If the funders had no role, please state: "The funders had no role in study design, data collection and analysis, decision to publish, or preparation of the manuscript.

Reviewers' comments:

Reviewer's Responses to Questions

**Comments to the Author**

1. Is the manuscript technically sound, and do the data support the conclusions?

Reviewer #1: Partly

Reviewer #2: Yes

2. Has the statistical analysis been performed appropriately and rigorously? 

Reviewer #1: No

Reviewer #2: I Don't Know

3. Have the authors made all data underlying the findings in their manuscript fully available?

Reviewer #1: Yes

Reviewer #2: Yes

4. Is the manuscript presented in an intelligible fashion and written in standard English?

Reviewer #1: No

Reviewer #2: Yes

5. Review Comments to the Author

Reviewer #1: Summary: This manuscript summarizes findings of an interesting analysis of predictors of viral non-suppression among men in fishing communities in Uganda, a high-risk population.

Major comments:

1. The outcome of interest of viral non-suppression should be defined. It is not clear what viral load threshold is being used in this outcome determination (i.e. not detected, 50 copies/mL, 1,000 copies/mL).

2. The multivariable model includes 15 predictors, many of which are multi-level categorical variables. Given that only 84 participants had non-suppression, there may be a risk of having overfit the model with too many predictors. The authors should provide justification for this or reconsider the variables that are included in the final model.

3. Would consider reframing conclusions to be more generalizable to a regional or global audience, rather than specific implications for the district health office for the area in which the study took place.

Minor comments:

1. The authors should use person-centered language in describing HIV status. Specifically, rather than referring to HIV positive adults, the authors should use the term “people with HIV.”

2. Methods – Study design: Further detail on the participant and healthworker interview regarding the health facilities would be helpful. Specifically, were interviews standardized by questionnaire, or were they open-ended and then coded using qualitative methodology? These methods and the factors explored should be more clearly explained.

3. Methods – Data collection: Not all factors explored in the regression analyses are explained in the methods section. Authors should note if adherence data are based on self-report, clinician assessment, or pill count. It would be also important to understand at what time point the predictors of interest of clinic and ART regimen are related to. For example, if ART regimen is listed as second-line, is that because a second-line regimen was prescribed at the visit when an elevated viral load was acted on, or was the viral load collected while on a second-line regimen? The same would be true for clinic, given that patients with virologic failure may be referred to higher level health centers.

4. Methods – Data analysis: It seems that a complete case analysis strategy was employed, rather than imputing missing data. If that is correct, this should be stated.

5. Results – When referring to 829 men in HIV care in Bulisa district, is this fully inclusive of all patients at all health facilities in the district, or only a subset? It would also be helpful to understand why 367 were excluded (which inclusion criteria were not met?).

6. Discussion – The authors note that longer duration between appointments was beneficial, but these data are not clearly located in the results table. Does this refer to the predictor labeled “clinical appointments and work”? The wording of the response options could be confused with referring to the length of the clinical visit itself, rather than the time interval between visits.

7. Discussion – The authors recommend the establishment of savings groups in fishing communities to improve treatment adherence. However, this isn’t discussed elsewhere in this analysis/manuscript.

8. Figure 1 – It would be helpful to be consistent with inclusion of percentages and numbers of participants.

9. Results – The authors should explain why age and regimen change were not included in the multivariable model.

10. The manuscript should be carefully reviewed and edited to correct typos, sentence structure errors, and spelling errors throughout.

Reviewer #2: Review of “Virological non-suppression among adult males attending HIV care services in the fishing communities in Bulisa district, Uganda”

Summary of review

Thank you for the opportunity to review this paper that outlined HIV virologic non-suppression and covariates. This is an important piece of work that adds to understanding one of the more vulnerable groups in Uganda. Overall, the manuscript was well-written. I have some specific points of clarification mainly in the Methods Section. Additionally, the Discussion section could be further expanded to better put the current findings into the broader research context.

Additional questions and suggestions are offered by manuscript section:

Overall

1. Please use people first language, for example instead of writing HIV positive, please change to people with HIV or men with HIV.

Introduction

1. It would be helpful for readers if you separated the one big Introduction paragraph into several smaller paragraphs.

2. The authors state that the HIV prevalence in the Bulisa district is 5.1%, but this is lower than the 2020 country-wide prevalence of 5.4% according to UNAIDS, please double-check the data and sources. It would be helpful to include years and sources in the text when feasible given that in this one sentence there appears to be comparisons among three different sources. It would also be helpful for the reader if you compared the Bulisa district HIV prevalence to the overall country-wide HIV prevalence for context.

“In a study carried out in Uganda using national programme data in 2017, the prevalence of nonsuppression among males was 13% [15]. However, in Bulisa district, with 14.2% of the population engaged in fishing business [16] and HIV prevalence of 5.1% [17], the

prevalence of virological non-suppression amongst all adult males for the same period was much higher at 20%.”

Methods

1. Under “study setting”, when describing the different management pathways for different patients, I would use the formal terminology – such as, “differentiated care models”.

2. Under “study setting”, please define facility based improved management.

3. Under “study design”, the first sentence is likely missing a word, “We conducted a cross-sectional *** utilising patients records and interviews to determine the prevalence of virological non-suppression and factors associated with virological nonsuppression among adult males, resident in the fishing communities of Bulisa district.”

4. Under “study design” please clarify whether interviews were also done with MWH participants.

5. Under “participant selection”, the following sentence is likely missing a word, “One health worker per facility was purposively selected for the face to *** interviews.”

6. Under “data collection and management”, please dscribe in more detail how adherence was categorized – was this self-report, a questionnaire, etc.?

7. Under “data analysis”, what was the definition of virologic non-suppression? What was the virologic threshold? What was the definition of “up to date” viral load – what was the viral load timing cut-off point for study inclusion?

8. Under “ethical considerations”, please also include if Uganda National Council of Science and Technology approval was obtained.

9. Under “data analysis”, please explain the rationale for when the authors chose to use means vs. medians. I would also outline that proportions and frequencies were described.

10. It would be helpful for the reader for a more detailed explanation of what the authors mean by landing sites – are these sites still in the same fishing villages – are they farther away? What are the implications of moving between different landing sites?

11. Please describe in greater details the questionnaire procedures with study staff. How were staff chosen to interview? How was the questionnaire designed?

Results

1. Please clarify over what amount of time participants reported number of sexual partners.

2. Under “Factors associated with virological non-suppression”, it would be more clear when comparing VL suppression in relation to income, if you compared the lower wage participants as baseline to the higher wage participants (so that your PR is >1 – similar to the other variables you have discussed). I would recommend this for any PR that is reported as <1 in the results.

Discussion

1. Paragraph 2, the authors state that men earning more money were less likely to have virological suppression. This appears to be the opposite of the data presented.

2. Consider “social desirability bias” to explain this concept – “Information bias could have also resulted from desirability by some participants responding to some questions like those assessing sexual behavior with reservations.”

3. I would recommend developing the first five or so paragraphs discussing the pertinent covariates of non-virologic suppression. For each point that you bring up, please expand the comparison to the literature and the implications for your specific research site.

4. An additional limitation could be the lack of back translation from the local language to English if this was not done.

Figures/Tables

1. Figure 1 has typos, please edit – ex: “Dint meet eligibility criteria 367”

6. PLOS authors have the option to publish the peer review history of their article (what does this mean?). If published, this will include your full peer review and any attached files.

Reviewer #1: No

Reviewer #2: No

---

## [Author Response · Author response to Decision Letter 0]

15 Feb 2023

Journal Requirements:

2. You indicated that you had ethical approval for your study. In your Methods section, please ensure you have also stated whether you obtained consent from parents or guardians of the minors included in the study or whether the research ethics committee or IRB specifically waived the need for their consent. Ascent from guardians of participants aged 15 to 18 years was obtained. 

>>>>>>This has been included under the methodology section. 

>>>>>>> This has been rectified.

4. Thank you for stating in your Funding Statement: "SI was partly funded by the Gilead foundation

BC was partly funded by the Fogarty International Centre, National Institute of Health (grant# 2D43TW009771-06 “HIV and co-infections in Uganda")

The management of people living with HIV was partly supported by the Presidents Emergency Plan for AIDS Relief through the United States Centers for Disease Control (CDC) and Prevention and the terms of cooperative agreement number NU2GGH001294-03-05" 

>>>>>> Funding statement amended

5. Thank you for stating the following financial disclosure: "SI was partly funded by the Gilead foundation

BC was partly funded by the Fogarty International Centre, National Institute of Health (grant# 2D43TW009771-06 “HIV and co-infections in Uganda")

The management of people living with HIV was partly supported by the Presidents Emergency Plan for AIDS Relief through the United States Centers for Disease Control (CDC) and Prevention and the terms of cooperative agreement number NU2GGH001294-03-05" 

>>>>>>>>>>>This has been rectified in the cover letter. 

Please state what role the funders took in the study. If the funders had no role, please state: "The funders had no role in study design, data collection and analysis, decision to publish, or preparation of the manuscript. If this statement is not correct you must amend it as needed. 

>>>>>>>> This has been included in the cover letter. 

>>>>>>> These have been included.

Reviewer #1: Summary: This manuscript summarizes findings of an interesting analysis of predictors of viral non-suppression among men in fishing communities in Uganda, a high-risk population.

Major comments: Comment well appreciated.

1. The outcome of interest of viral non-suppression should be defined. It is not clear what viral load threshold is being used in this outcome determination (i.e. not detected, 50 copies/mL, 1,000 copies/mL). >>>>>>>> Addressed in the edited document.

2. The multivariable model includes 15 predictors, many of which are multi-level categorical variables. Given that only 84 participants had non-suppression, there may be a risk of having overfit the model with too many predictors. The authors should provide justification for this or reconsider the variables that are included in the final model. 

>>>>>>>Major edits were done following this comment. 88 participants had virological non-suppression. To avoid the risk of overfitting the model as guided, the 1 in 10 to rule was used and only 9 independent variables were used. (8.8 rounded off to 9). Several variables in the previous model were dropped and some were replaced in the final model. 

3. Would consider reframing conclusions to be more generalizable to a regional or global audience, rather than specific implications for the district health office for the area in which the study took place.

>>>>>> Conclusions and recommendations have been reframed as guided.

Minor comments:

1. The authors should use person-centered language in describing HIV status. Specifically, rather than referring to HIV positive adults, the authors should use the term “people with HIV.” 

>>>>>>> This was addressed throughout the whole document.

2. Methods – Study design: Further detail on the participant and healthworker interview regarding the health facilities would be helpful. Specifically, were interviews standardized by questionnaire, or were they open-ended and then coded using qualitative methodology? These methods and the factors explored should be more clearly explained. 

>>>>>>> Exploration on the factors considered was done. 

>>>>>>> Elaboration on the interviewing process included. 

3. Methods – Data collection: Not all factors explored in the regression analyses are explained in the methods section. Authors should note if adherence data are based on self-report, clinician assessment, or pill count. It would be also important to understand at what time point the predictors of interest of clinic and ART regimen are related to. For example, if ART regimen is listed as second-line, is that because a second-line regimen was prescribed at the visit when an elevated viral load was acted on, or was the viral load collected while on a second-line regimen? The same would be true for clinic, given that patients with virologic failure may be referred to higher level health centers.

>>>>>> All factors explored in the regression analysis have been explained in the methodology section A viral load within a period of 12 months from the time of sample collection was compiled for this study. The relationship of this viral load with the duration on second line regimen was compiled for this study. So, it is hard to tell whether a high viral load was related to deciding to switch to second line or it applied to a patient who had been on second line for atleast 6 month. 

This is one of the indicators that were replaced while reducing the number of variables at multivariable analysis

4. Methods – Data analysis: It seems that a complete case analysis strategy was employed, rather than imputing missing data. If that is correct, this should be stated. 

>>>>>> Case based analysis was used

5. Results – When referring to 829 men in HIV care in Bulisa district, is this fully inclusive of all patients at all health facilities in the district, or only a subset? It would also be helpful to understand why 367 were excluded (which inclusion criteria were not met?). 

>>>>> Inclusion criteria was included in the consort diagram.

6. Discussion – The authors note that longer duration between appointments was beneficial, but these data are not clearly located in the results table. Does this refer to the predictor labeled “clinical appointments and work”? The wording of the response options could be confused with referring to the length of the clinical visit itself, rather than the time interval between visits. 

>>>>>> Clinical appointments were assessed as long or short according to the participants perception.

Time interval between visits is much better as suggested for consideration under discussion. 

7. Discussion – The authors recommend the establishment of savings groups in fishing communities to improve treatment adherence. However, this isn’t discussed elsewhere in this analysis/manuscript. 

>>>>>>> Saving groups had been suggested as a way to boost the income of participants since average income had an association with virological non-suppression. However, the guidance on the wording is noted for consideration.

8. Figure 1 – It would be helpful to be consistent with inclusion of percentages and numbers of participants. >>>>> All percentages have been included in the table

9. Results – The authors should explain why age and regimen change were not included in the multivariable model. 

>>>>> Age has been included in the revised model as one of the universal confounders. 

>>>>> At Multivariable analysis, inclusion of variables was mainly determined by the uniqueness in relation to fishing communities. Variables with a p-value less than 0.25 were included and eliminated until a better model was obtained. Regimen change was one of them. Multicollinearity is also another reason why it was eliminated. 

10. The manuscript should be carefully reviewed and edited to correct typos, sentence structure errors, and spelling errors throughout. 

Reviewer #2: Review of “Virological non-suppression among adult males attending HIV care services in the fishing communities in Bulisa district, Uganda”

Summary of review

Thank you for the opportunity to review this paper that outlined HIV virologic non-suppression and covariates. This is an important piece of work that adds to understanding one of the more vulnerable groups in Uganda. Overall, the manuscript was well-written. I have some specific points of clarification mainly in the Methods Section. Additionally, the Discussion section could be further expanded to better put the current findings into the broader research context.

Additional questions and suggestions are offered by manuscript section:

Overall

1. Please use people first language, for example instead of writing HIV positive, please change to people with HIV or men with HIV.

 This was addressed throughout the whole document.

Introduction

1. It would be helpful for readers if you separated the one big Introduction paragraph into several smaller paragraphs. >>>>>> This was done

2. The authors state that the HIV prevalence in the Bulisa district is 5.1%, but this is lower than the 2020 country-wide prevalence of 5.4% according to UNAIDS, please double-check the data and sources. It would be helpful to include years and sources in the text when feasible given that in this one sentence there appears to be comparisons among three different sources. It would also be helpful for the reader if you compared the Bulisa district HIV prevalence to the overall country-wide HIV prevalence for context.

 >>>>>> This was rectified

“In a study carried out in Uganda using national programme data in 2017, the prevalence of non-suppression among males was 13% [15]. However, in Bulisa district, with 14.2% of the population engaged in fishing business [16] and HIV prevalence of 5.1% [17], the prevalence of virological non-suppression amongst all adult males for the same period was much higher at 20%.”

 >>>>> This was rectified

Methods

1. Under “study setting”, when describing the different management pathways for different patients, I would use the formal terminology – such as, “differentiated care models”. >>>> Adopted as guided

2. Under “study setting”, please define facility based improved management. >>>> This was defined

3. Under “study design”, the first sentence is likely missing a word, “We conducted a cross-sectional *** utilising patients records and interviews to determine the prevalence of virological non-suppression and factors associated with virological nonsuppression among adult males, resident in the fishing communities of Bulisa district.”

 >>>>> This was rectified

4. Under “study design” please clarify whether interviews were also done with MWH participants.

 >>>> MWH not clear to me. But interview section was rectified

5. Under “participant selection”, the following sentence is likely missing a word, “One health worker per facility was purposively selected for the face to *** interviews.” >>>> This was rectified

6. Under “data collection and management”, please dscribe in more detail how adherence was categorized – was this self-report, a questionnaire, etc.? >>>> Categorization of adherence was included

7. Under “data analysis”, what was the definition of virologic non-suppression? What was the virologic threshold? What was the definition of “up to date” viral load – what was the viral load timing cut-off point for study inclusion? >>>>> All this has been clarified under data collection and management. 

8. Under “ethical considerations”, please also include if Uganda National Council of Science and Technology approval was obtained. >>>>> This study was done as part of the fulfilments required for the awards of Master’s degree in Public health at Makerere University. 

The university/ IRB has the responsibility to submit a list of student’s researches to UNCST. My letter for approval is attached here. 

9. Under “data analysis”, please explain the rationale for when the authors chose to use means vs. medians. I would also outline that proportions and frequencies were described. 

 >>>>> Mean (SD) was used as a measure of central tendency for age since this had a normal distribution. Duration on ART had a skewed distribution hence median (IQR) was used as the measure of central tendency.

10. It would be helpful for the reader for a more detailed explanation of what the authors mean by landing sites – are these sites still in the same fishing villages – are they farther away? What are the implications of moving between different landing sites? >>>> Landing sites have been defined under study setting.

The phrase “landing sites” has been edited to “fish landing sites”

11. Please describe in greater details the questionnaire procedures with study staff. How were staff chosen to interview? How was the questionnaire designed?

 >>>This has been addressed under study design and participants selection.

Results

1. Please clarify over what amount of time participants reported number of sexual partners. >>>> This has been addressed

2. Under “Factors associated with virological non-suppression”, it would be more clear when comparing VL suppression in relation to income, if you compared the lower wage participants as baseline to the higher wage participants (so that your PR is >1 – similar to the other variables you have discussed). I would recommend this for any PR that is reported as <1 in the results.

 >>>> This was considered for all other variables that has PR < 1 included at multivariable analysis.

Discussion

1. Paragraph 2, the authors state that men earning more money were less likely to have virological suppression. This appears to be the opposite of the data presented.

 >>>> Average income is one of the variables that were dropped when cutting down the number of variables from 15 to 9 at multivariable analysis. Justification for cutting down the variables was considered to avoid the risk of over fitting the model following the guidance from reviewer one but also from the reviewed literature. 

2. Consider “social desirability bias” to explain this concept – “Information bias could have also resulted from desirability by some participants responding to some questions like those assessing sexual behavior with reservations.” >>> Indeed, social desirability explains the responses to sexual behavior questions better as guided. This has been considered in the manuscript. 

3. I would recommend developing the first five or so paragraphs discussing the pertinent covariates of non-virologic suppression. For each point that you bring up, please expand the comparison to the literature and the implications for your specific research site. >>>>>> This was rectified under discussion

4. An additional limitation could be the lack of back translation from the local language to English if this was not done.

 >>>>> This has been considered.

Figures/Tables

1. Figure 1 has typos, please edit – ex: “Dint meet eligibility criteria 367” >>>> Typos have been revised in the entire document

6. PLOS authors have the option to publish the peer review history of their article (what does this mean?). If published, this will include your full peer review and any attached files.

No objection

---

## [Decision Letter · Decision Letter 1]

11 Apr 2023

PONE-D-21-31809R1Virological non-suppression among adult males attending HIV care services in the fishing communities in Bulisa district, Uganda.PLOS ONE

Dear Dr. Senteza,

Thank you for submitting your manuscript to PLOS ONE. After careful consideration, we feel that it has merit but does not fully meet PLOS ONE’s publication criteria as it currently stands. Therefore, we invite you to submit a revised version of the manuscript that addresses the points raised during the review process.

We look forward to receiving your revised manuscript.

Kind regards,

Joel Msafiri Francis, MD, MS, PhD

Academic Editor

PLOS ONE

Journal Requirements:

Reviewers' comments:

Reviewer's Responses to Questions

**Comments to the Author**

1. If the authors have adequately addressed your comments raised in a previous round of review and you feel that this manuscript is now acceptable for publication, you may indicate that here to bypass the “Comments to the Author” section, enter your conflict of interest statement in the “Confidential to Editor” section, and submit your "Accept" recommendation.

Reviewer #2: All comments have been addressed

2. Is the manuscript technically sound, and do the data support the conclusions?

Reviewer #2: Yes

3. Has the statistical analysis been performed appropriately and rigorously? 

Reviewer #2: I Don't Know

4. Have the authors made all data underlying the findings in their manuscript fully available?

Reviewer #2: Yes

5. Is the manuscript presented in an intelligible fashion and written in standard English?

Reviewer #2: Yes

6. Review Comments to the Author

Reviewer #2: Summary of review

The authors have done a commendable job incorporating the previous recommendations. I have additional recommendations to further strengthen the manuscript outlined below.

Overall

1. There continue to be typos and English grammar problems throughout the manuscript.

2. Please use precise writing throughout your manuscript – several examples of imprecision are outlined below in the Intro paragraph, “which district”, “prevalence of what”?

3. I’m struggling to understand how the health center survey fits in with the virologic suppression analysis. A starting point would be to describe the clinic/hospital level of each health center you surveyed. Is there a way to tie in the six health centers into the multivariable analysis of viral suppression?

Introduction

1. Page 3, Para 2 - This sentence is grammatically incorrect – please revise, “This mix of potential drivers of HIV transmission rises the need to contextualize more efficient methods of HIV prevention.

2. Page 3, Para 2 – “Basing on the assertion that undetectable viral load translates into no transmission of HIV (undetectable = untransmittable) [9-12], understanding the determinants of virological suppression could help in reducing the incidence of HIV in fishing communities [13, 14].” Change basing to based.

3. Overall, Paragraph 3 is confusing. I am not understanding the comparisons or what we are always discussing due to imprecise language.

4. Page 3, Para 3 – What are you comparing – what is this higher than? “The prevalence of HIV amongst the residents of the fishing communities is not known but is likely to be higher given the observed prevalence of HIV in the fishing communities along Lake Victoria[5].”

5. Page 3, Para 3 – clarify which district, “However, the general prevalence of HIV in the district based on 2020 program data in DHIS-2 was lower at 5.1% [16] as compared to the national prevalence of 6.2% reported in the UPHIA report of 2017 [6].”

6. Page 3, Para 3 – These percentages of non-suppression are among men with HIV (MWH) correct? As written, you state that this percentage is among all adult men. “In a study carried out in Uganda using national programme data in 2017, the prevalence of non-suppression among males was 13%. However, in Bulisa district, the prevalence of virological non-suppression amongst all adult males for the same period was much higher at 20% [17].”

7. This doesn’t make sense – you’re comparing the same thing, “Whereas there is the availability of program data about virological non-suppression for the districts with fishing communities in Uganda, little is known about the burden of virological non-suppression among the male-dominated fishing communities.

Methods

1. Page 4, last sentence of “Study Setting” – what is the difference between facility-based management and facility-based improved management model, “40% on facility-based management, 6% on client-led ART delivery and 3% on the facility based improved management model.”

2. Under “Participant selection”, Is the “in charge” a person? Can you provide more detail (ie the nurse in charge)? “The HIV clinic in charge at each of the six health facilities serving patients living with HIV in the fishing communities was purposively selected for the face-to-face interviews to assess the contribution of the health facility to virological non-suppression.”

3. Page 6, “Data Collection Management”, I think you might mean 12 months from this study? As written it sounds like you mean the viral load was run up to 12 months after that blood vial’s collection. “An up-to-date viral load result was defined as a viral load result within 12 months from the time of taking off the blood sample”

4. Page 7, was pill count adherence based on participant report or did research staff count participant pills?

Results

1. Page 10, It would be helpful to clarify the corresponding age with each group. “However, they differed in their mean (SD) age (years), 40 (10.7) versus 35.9 (9.3), p=0.011.”

2. Page 10, It makes more sense to compare the proportion of participants with viral non-suppression to those who did not did not participate in the study instead of comparing VL nonsuppression among participants with any eligible participant.

3. Page 10, under “Demogrpahics” please define “mid-level health facility”.

4. Page 10, under “Demographics”, “About one-third reported having had 2 to 3 sexual partners in the previous 12 months and one-third never knew their partner’s HIV status.” Is partner knowledge status in reference to all partners? Any one partner? Please clarify.

5. Page 13-14, under “Bivariable and multivariable analysis, you only discuss Bivariable analysis. I would potentially shorten the bivariable results and give more space to highlight the big takeaways from the multivariable analysis.

Discussion

1. I suggest separating the discussion into cohesive paragraphs. I suggest reading other Discussion sections from high-impact journals for reference. A typical discussion paragraph will take a finding from your study that’s interesting to you, and put it in the context of broader research. You’ll typically try to compare this finding to research in similar settings or populations. Typically, you’ll want to describe the other studies a bit as your reader won’t necessarily have time to look at your references. You’ll then try to formulate a take-away, such as a suggestion or interpretation for how your study finding fits into the larger data. You don’t need to address every finding from your results section – only the findings you think are interesting. You begin to do this later in the discussion section.

2. I would try to better tie in the health center survey to the virologic nonsuppression outcome. As it stands these seem like two completely separate projects. Better framing in the discussion could help this issue.

Figures/Tables

1. Table 1. Under “number of sexual partners in previous 12 mo”, I suggest changing “only one” to “one”.

2. Table 1. Under “Quality of clinical appointments and work”, what is the difference between “long enough” and “they are just fine”? These two fields do not seem exclusionary.

3. Table 2. I am confused by the column headings: “No – suppressed” and Yes-non-suppressed”. I would remove the No/Yes and simply write suppressed vs. non-suppressed.

4. Table 2. It looks like you are trying to bold any p-values <0.05. If this is the case, you should also bold TB diagnosed in past 12 mo.

5. Table 3 took me a few minutes to comprehend. It might be visually easier to digest the information you’re trying to describe by having the six health centers at column headers, and then simply checkmark which health centers respond yes to each question.

7. PLOS authors have the option to publish the peer review history of their article (what does this mean?). If published, this will include your full peer review and any attached files.

Reviewer #2: No

---

## [Author Response · Author response to Decision Letter 1]

29 Jun 2023

We appreciate the support provided by the reviewers in making our work better. We are really grateful.

All comments suggested were of great importance and we believe we have presented a better version of the manuscript.

---

## [Decision Letter · Decision Letter 2]

19 Jul 2023

PONE-D-21-31809R2Virological non-suppression among adult males attending HIV care services in the fishing communities in Bulisa district, Uganda.PLOS ONE

Dear Dr. Senteza,

Thank you for submitting your manuscript to PLOS ONE. After careful consideration, we feel that it has merit but does not fully meet PLOS ONE’s publication criteria as it currently stands. Therefore, we invite you to submit a revised version of the manuscript that addresses the points raised during the review process.

We look forward to receiving your revised manuscript.

Kind regards,

Joel Msafiri Francis, MD, MS, PhD

Academic Editor

PLOS ONE

Journal Requirements:

Reviewers' comments:

Reviewer's Responses to Questions

**Comments to the Author**

1. If the authors have adequately addressed your comments raised in a previous round of review and you feel that this manuscript is now acceptable for publication, you may indicate that here to bypass the “Comments to the Author” section, enter your conflict of interest statement in the “Confidential to Editor” section, and submit your "Accept" recommendation.

Reviewer #2: (No Response)

2. Is the manuscript technically sound, and do the data support the conclusions?

Reviewer #2: Yes

3. Has the statistical analysis been performed appropriately and rigorously? 

Reviewer #2: I Don't Know

4. Have the authors made all data underlying the findings in their manuscript fully available?

Reviewer #2: Yes

5. Is the manuscript presented in an intelligible fashion and written in standard English?

Reviewer #2: Yes

6. Review Comments to the Author

Reviewer #2: Overall

The authors have done a commendable job incorporating the previous recommendations. I have additional recommendations to further strengthen the manuscript outlined below.

Introduction

1. Page 3, Paragraph 1: A prevalence is not a rate – please revise.

2. Page 4, Paragraph 1, Line 1: Clarify type of non-suppression (virologic) as done later in the sentence.

Methods

1. Study setting: I recommend including the number of people living in this district/setting – this would put the number of people living with HIV into context.

2. Page 7, Data collection and management: Please cite references for the sentences, The patients’ questionnaire was structured and designed based on characteristics common among residents of fishing communities and factors previously associated with virological non-suppression from other population segments” and “The Health worker’s questionnaire was structured and designed based on factors identified

in previous qualitative and quantitative studies that focused on the quality of health care.”

3. Page 7, punctuation errors in the phrase: “adherence based on pill count by the attending clinician categorized as good (>95%), fair (80-95%) poor (<80%), fulfilment of clinical appointments;”should be, “adherence based on pill count by the attending clinician categorized as good (>95%), fair (80-95%), poor (<80%); fulfilment of clinical appointments;”

4. Page 10, The following sentence does not make sense, “Out of the 462 participants who fulfilled the eligibility criteria, 367 were excluded because they were non-residents”. Residing in the district is an eligibility criteria and above the authors state 462 participants met criteria.

5. Page 11, the authors compare participants interviewed vs. those who did were not interviewed for several categories, but they compare participants interviewed to all participants for viral load. I recommend they consistently compare those interviewed vs. those who did not interview.

6. How did the authors ask about HIV disclosure? Could participants state disclosure to multiple categories or were they to only choose one? Please clarify in the manuscript.

7. How did the authors partner HIV status in regards to people with multiple partners? Please clarify in the manuscript.

8. Under quality of clinical appointments and work, please clarify the difference between, “long enough” and “they are just fine”. Please clarify in the manuscript.

9. Under viral load turn around time, please clarify “can’t tell”. Does this mean the participant didn’t know or didn’t want to report this in the study?

Results

1. Please define CPR at first use.

2. It is difficult to think about prevalence ratios < 1. I recommend reversing your reference labels to have results where your prevalence ratios are all > 1.

3. Please revise this sentence as it is difficult to follow, “The prevalence of virological non-suppression amongst participants who received care from the only hospital in the district was 1.22 times (adj.PR=1.22, 95%CI: 1.08-1.37) and 0.61 (adj.PR=0.61, 95%CI: 0.57-0.66) amongst participants that sought treatment at health centre III levels compared to the prevalence of virological non-suppression amongst participants who received care at health centre II level.”

4. Overall you could cut down significantly on the word count of the Results section. Since this data is in the Tables, you should only write out the most important findings you wish to highlight.

5. Why wasn’t ARV regimen change or ARV line of therapy in the multivariable analysis since these seemed to significantly associated with virologic nonsuppression?

6. It appears that there are 9 variables in the multivariable model. Are the authors concerned about overfitting the model? Especially since participants without viral suppression were 84 (in regards to the 1/10 regression rule).

7. The data regarding the health facility factors does not seem relevant in this analysis since the authors do not tie this into the viral nonsuppression in any way. I would recommend removing this and potentially it could be a short report.

Discussion

1. In regards to limitations, the authors state, “An additional limitation could be the lack of back translation from the local language to English which could have led to a loss of meaning in the translation of some data.” If the authors and people doing the analysis understood and were fluent in the local language, then there is no need for English translation.

Figures/Tables

1. Table 3. Please remove the “No” row for each question. This is confusing. The “Yes” row alone is sufficient.

7. PLOS authors have the option to publish the peer review history of their article (what does this mean?). If published, this will include your full peer review and any attached files.

Reviewer #2: No

---

## [Author Response · Author response to Decision Letter 2]

3 Oct 2023

These have been included in the attachment labelled response to reviewers.

---

## [Editor Report · Decision Letter 3]

5 Oct 2023

Virological non-suppression among adult males attending HIV care services in the fishing communities in Bulisa district, Uganda.

PONE-D-21-31809R3

Dear Dr. Senteza,

We’re pleased to inform you that your manuscript has been judged scientifically suitable for publication and will be formally accepted for publication once it meets all outstanding technical requirements.

Kind regards,

Joel Msafiri Francis, MD, MS, PhD

Academic Editor

PLOS ONE
---

## [Editor Report · Acceptance letter]

10 Oct 2023

PONE-D-21-31809R3 

Virological non-suppression among adult males attending HIV care services in the fishing communities in Bulisa district, Uganda. 

Dear Dr. Senteza:

I'm pleased to inform you that your manuscript has been deemed suitable for publication in PLOS ONE. Congratulations! Your manuscript is now with our production department. 

Kind regards, 

on behalf of

Dr. Joel Msafiri Francis 

Academic Editor

PLOS ONE